# Methadone and Buprenorphine as Medication for Addiction Treatment Diversely Affect Inflammation and Craving Depending on Their Doses

**DOI:** 10.3390/pharmacy13020040

**Published:** 2025-03-06

**Authors:** Christonikos Leventelis, Aristidis S. Veskoukis, Andrea Paola Rojas Gil, Panagiotis Papadopoulos, Maria Garderi, Asimina Angeli, Antzouletta Kampitsi, Maria Tsironi

**Affiliations:** 1Department of Nursing, University of Peloponnese, 22100 Tripoli, Greece; arojas@uop.gr (A.P.R.G.); mtsironi@otenet.gr (M.T.); 2Organization Against Drugs, 10433 Athens, Greece; panpap1973@yahoo.gr (P.P.); mariagarderi@gmail.com (M.G.); asiminaangeli@yahoo.gr (A.A.); 3Department of Nutrition and Dietetics, University of Thessaly, Argonafton 1, 42132 Trikala, Greece; veskoukis@uth.gr; 4General Anticancer Hospital “Agios Savvas”, 11522 Athens, Greece; anjuletta@yahoo.gr

**Keywords:** methadone, buprenorphine, opioid maintenance treatment, inflammation, growth factors, cytokines

## Abstract

Buprenorphine and methadone are widely used as medication for addiction treatment (MAT) in patients with opioid use disorders. However, there is no compelling evidence of their impact on the immune–endocrine response. Therefore, the aim of this study was to examine the effects of the aforementioned medications on craving and on biomarkers of inflammation and cortisol, approaching the dose issue concurrently. Sixty-six patients (thirty-four under methadone and thirty-two under buprenorphine) who had just entered a MAT program and were stabilized with the suitable administered doses after a two-week process were divided into four groups based on medication dose (i.e., methadone high dose, buprenorphine high dose, methadone medium dose, and buprenorphine medium dose). The heroin craving questionnaire for craving assessment was completed, and the blood biomarkers were measured on Days 1 and 180. According to the results, high doses of both medications were accompanied by low levels of craving, cortisol, and inflammation on Day 1, and no alterations were observed on Day 180. On the contrary, medium doses reduced the tested psychosocial and biochemical parameters in terms of time, indicating a positive action for the patients. Concludingly, modifications in MAT doses are needed soon after the stabilization process to prevent inflammation and avoid relapse, thus helping opioid-addicted patients toward rehabilitation.

## 1. Introduction

Opioid use disorders (OUDs) refer to an overpowering desire for opioid use and an increased tolerance along with withdrawal and addiction manifested by neurobiological, behavioral, and psychological alterations [1]. Opioids act at the presynaptic nerve terminal by inhibiting dopamine release and at the postsynaptic neuron by blocking mu opioid receptors (MORs), hence preventing the secretion of γ-aminobutyric acid (GABA). GABA acts on dopaminergic neurons by inhibiting dopamine release [2,3]. The action of opioids on GABA is associated with the reward system. This is a critical factor in the further development of relapsing disorder, leading to craving and drug-seeking behavior [2,3].

Methadone as methadone maintenance treatment (MMT) and buprenorphine as buprenorphine maintenance treatment (BMT) are the basic kinds of medication for addiction treatment (MAT). At the biochemical level, methadone binds to MORs as a full agonist and activates both kappa opioid receptors (KORs) and delta opioid receptors (DORs), causing mild withdrawal symptoms [4,5]. Buprenorphine is a partial agonist of MORs and an antagonist of KORs, displaying a ceiling effect, thus inducing fewer euphoric symptoms [5,6]. Apart from the attenuation of serious health abnormalities, it is evident that MMT and BMT reduce the transmission of serious diseases, including acquired immunodeficiency syndrome (AIDS) and hepatitis [7,8]. Moreover, they potentially decrease the use of illicit substances, thus contributing to the improvement of the quality of life of the patients with OUDs [9]. Even though MAT is considered beneficial for patients, it has been associated with severe side effects on neuro-endocrine mechanisms associated with oxidative stress [5,10] and neuro-inflammation, hence strengthening drug-seeking behaviors and craving [11,12].

Craving, as an urge to use illicit substances, is a stressful and crucial symptom related to abnormal neuro-immuno-endocrine interactions, resulting in the activation of the hypothalamic–pituitary–adrenal (HPA) axis [13]. This process is followed by cortisol, the so-called hormone of stress, secretion with a parallel immune response by pro-inflammatory mediators and cytokine production [13,14]. Recent findings have indicated that opioid use, due to craving and the abnormal function of the HPA axis, is related to the upregulation of pro-inflammatory cytokines, such as tumor necrosis factor-alpha (TNF-α), C-reactive protein (CRP), interleukin-6 (IL-6), and interleukin-1 beta (IL-1β) [14,15]. According to the notion supported by the literature, different opioids, especially long-acting ones such as methadone and buprenorphine, do not share the same immunomodulatory effects [16,17].

A limited number of studies have revealed the effects of MMT or BMT on the neuro-immune system interaction in association with craving. It has been observed that the effects of mu opioid agonists on the immune system may be mediated by indirect activation of the HPA axis, increased cortisol secretion, and release of pro-inflammatory agents [16]. Furthermore, preclinical studies have indicated that MMT was less effective in macrophage activation and antibody generation than other opioids, suggesting that methadone is related to a weak immunological response [18,19]. In addition, in vitro studies in human cell lines have shown a significant decrease in IL-6 production by T-lymphocytes due to methadone [20]. Similarly, buprenorphine also appears to be effective in immune response since no NK lymphocyte activation or differentiation in macrophage function was reported [21,22], whereas it improved immune parameters and preserved immune function in vivo [23]. However, further evidence has suggested that mu opioid agonists, such as methadone, induce pro-inflammatory activity in the central nervous system through reactive oxygen species (ROS) production [24]. In addition, human studies have revealed that methadone exerted an immunoregulatory function on OUD patients who have been infected by hepatitis C and HIV [25,26], with the upregulation of cytokine 10 (IL-10) controlling inflammation [27]. Moreover, buprenorphine may reduce inflammatory biomarkers by inhibiting adherence of monocytes to surface receptors in OUD patients suffering from HIV [28,29]. However, it has also been found that higher levels of IL-6 are associated with increased dropout rates in MMT patients [30], increased ROS generation in both MMT and BMT patients, and elevated cortisol levels due to the downregulation of the HPA axis [5,31].

Based on the above, it appears that MAT helps patients partly reduce the organismal damage due to opioid abuse. However, the biochemical role of methadone and buprenorphine on the response of neuro-endocrine and immune systems remains unclear. Therefore, the main objective of the present investigation was to evaluate the effects of methadone and buprenorphine on craving, the primary parameter that needs to be improved toward rehabilitation, and on the response of neuro-endocrine and immune systems through the assessment of inflammation biomarkers. The issue of the MAT dose on the aforementioned biochemical pathways is also approached, giving this investigation enhanced clinical importance.

## 2. Materials and Methods

### 2.1. Participants

The participants were patients with OUDs attending MAT programs. A statistical power analysis with the online sample size calculator “ClinCalc.com (assessed on 12 May 2023)” as performed to enroll an adequate number of participants in this study. The power analysis was based on the reduction of craving as the primary outcome. According to the literature, an approximate 40% reduction in craving has been observed in the higher dose of MAT [32]. On that basis, the power analysis revealed that at least 29 participants for each group (i.e., MMT and BMT) are needed for statistically meaningful results. The input variables for the analysis were as follows: alpha error = 0.05, power = 0.80, and enrollment ratio = 1. Due to a potential 10% dropout, the final number of participants was n = 32 for the BMT group and n = 34 for the experimental group.

The total number of the participants (n = 66) was further randomly divided using the NCI (National Cancer Institute) clinical trial randomization tool [33]. The randomization process, which was based on the patient’s unique code when entering the MAT programs and administering medication, was performed by an independent individual who did not participate in the data collection and evaluation procedure. This individual had access to interim results and was responsible for terminating the procedure in case there was a need for that or any adverse effects had been raised before being addressed to the appropriate expert. According to the aforementioned randomization process and power analysis, the volunteers were divided into two independent groups discriminated by medication dose, named herein as the group under medium dose (MD, n = 32) and the group under high dose (HD, n = 34) for each administered substitute (i.e., methadone and buprenorphine) forming 4 groups: group A (methadone medium dose—MMD, n = 17), group B (buprenorphine medium dose—BMD, n = 15), group C (methadone high dose—MHD, n = 17), and group D (buprenorphine high dose—BHD, n = 17). This stratification was carried out as follows: for the MMT patients, the medium dose ranged between 45 and 85 mg, and high dose was considered a dose > 85 mg [34]. The BMT patients were similarly stratified in medium doses ranging between 8 and 16 mg and high doses, namely > 16 mg and lower than 45 mg [35]. The volunteers were fully informed about the purpose and objectives of the study. All necessary information and safeguards were given to ensure the confidentiality of data, and each patient signed a consent form for participation and data publication before this study began. According to the inclusion criteria, the patients should be over 20 years of age under opioid substance addiction and, thus, suffering from physical and mental dependence due to opioid use. Patients with severe psychopathology and serious medical problems, including human immunodeficiency virus (HIV), hepatitis B and C, or under anti-inflammatory treatment, were excluded because such conditions create difficulties in the program monitoring. Patients with relapse to other substances (i.e., opioids, methamphetamine, amphetamine, benzodiazepines, tetrahydrocannabinol, and cocaine) were also excluded from this study. If any participants should change the medication dose, they would be excluded as well. None of the patients who participated were found to be in this situation.

### 2.2. Experimental Design

All patients had just entered the MAT program and were stabilized with the suitable administered doses of methadone and buprenorphine after the appropriate process (approximately 2 weeks), according to WHO guidelines [36]. The experimental procedure lasted for 6 months. Collection of blood samples for the measurement of biomarkers of inflammation was carried out early in the morning before the medication administration at two time points. In particular, blood samples were drawn on Day 1 or baseline (at the start of the experiment, i.e., the first day after stabilization procedure) and Day 180 (at the end of the experiment). The Heroin Craving Questionnaire (HCQ) was also completed by the patients at the same time points to estimate craving. To rule out the use of other substances (i.e., opioids, methamphetamine, amphetamine, benzodiazepines, tetrahydrocannabinol, and cocaine), a weekly urine test was performed on all participants. All patients were found negative for substance abuse. Demographic data, including age, gender, area of residence, years attending MAT, age started using, and duration of using addictive substances before entering a MAT program, were obtained.

### 2.3. Drug Administration

Methadone hydrochloride solution (10 mg/ml) and buprenorphine/buprenorphine-naloxone pills (2 mg–8 mg) were used. The moderate methadone mean daily dose was approximately 60 mg/24 h, and the high mean daily dose was 105 mg/24 h. Similarly, the moderate mean daily dose of buprenorphine was equal to 12 mg, and the mean high dose was 20 mg/day.

### 2.4. Craving Assessment

The validated HCQ was used for the assessment of craving comprising five items, namely *desire to use heroin*, *intentions and planning to use heroin*, *anticipation of a positive outcome*, *relief from withdrawal or dysphoria*, and *lack of control over use*; each of them consists of nine questions. The score of each dimension is calculated with the use of a 7-point Likert scale ranging from 1 (i.e., strongly disagree) to 7 (i.e., strongly agree), resulting in a total craving score, wherein the higher the number, the higher the level of craving. The questionnaire is a reliable instrument with Cronbach’s alpha equal to 0.90 [37].

### 2.5. Biochemical Measurements in Blood

Biomarkers of inflammation, namely the pro-inflammatory interferon gamma (IFNγ), interferon alpha-2 (IFNa-2), interleukin-1 beta (IL-1β), interleukin-8 (IL-8), and monocyte chemoattractant protein-1 (MCP-1), were measured. In addition, cortisol and the anti-inflammatory interleukin-10 (IL-10), as well as epidermal growth factor (EGF), basic fibroblast growth factor (FGF-2), and transforming growth factor-alpha (TGF-a), were measured as well. IFNγ is a cytokine that recruits macrophages at the site of inflammation [38], whereas IFNa2 regulates the immune system activation and inhibits cell proliferation [39]. IL-1 alpha and IL-1 beta are activated mainly by macrophages and neutrophils regulating immune response [40], while IL-8, as a chemokine, induces chemotaxis of neutrophils and other granulocytes to an inflammation site [41]. MCP-1 belongs to the family of chemokines and recruits monocytes, lymphocytes, and neutrophils at the inflammation site [42,43]. Moreover, IL-10 acts protectively against inflammation since it increases immunoglobulin secretion and downregulates the activity of several cytokines [44]. TGF-a, FGF-2, and EGF belong to growth factors contributing to proliferation, growth, cell differentiation, and cell signal transmission [45,46]. All biomarkers were measured in both time points (i.e., Days 1 and 180). Blood samples were drawn from a forearm vein and stored in clotting tubes. The blood was clotted for 30 min before centrifugation (10 min, 1000× *g*, 4 °C). The supernatant (i.e., the serum) was removed and used for the measurement of cortisol levels (Cayman Cortisol Elisa Kit, Cayman Chemical, 1180 East Ellsworth Road, Ann Arbor, MI 48108 USA) and the aforementioned cytokines and growth factors with a Milliplex Human Cytokine/Chemokine Magnetic Bead Panel (Merck Millipore, Worldwide Headquarters, 400 Summit Drive, Burlington, MA, USA) according to the manufacturer’s instructions and detected with a multiplex detection platform (Luminex^®^ 200™ System).

### 2.6. Ethical Approval

All applied experimental procedures were in accordance with the European Union Guidelines laid down in the 1964 Declaration of Helsinki, as revised in 2013. They were also approved by the Institutional Review Board of the University of Peloponnese (Tripoli, Greece) and the Organization Against Drugs (ref. number 5426/15-4-2012).

### 2.7. Statistical Analysis

Continuous variables are presented as mean ± standard deviation (SD). The alterations of the levels of craving, as well as the concentrations of the biomarkers measured in blood, were assessed through two-way, 2 × 2 (i.e., MAT dose × time), analysis of variance (ANOVA) with repeated measures. The results from the parameters of the craving questionnaire were correlated with the results from the biochemical parameters through Pearson’s correlation test (r-value). The statistical significance was set at *p* < 0.05. All analyses were conducted using SPSS statistical software (version 22.0).

## 3. Results

### 3.1. Demographic Characteristics

The characteristics of the participants as a total and per group are presented in Table 1. All groups were found to be similar in terms of all demographic parameters, as indicated by the statistical analysis.

### 3.2. Craving

The effects of MAT doses on craving through dimensions of HCQ are presented in Table 2. According to the results for the MMT patients under a high MAT dose on Day 1, a significantly lower score in the dimensions of *desire to use heroin* (*p* = 0.04) and *intention and planning to use* (*p* = 0.04) was observed compared to their counterparts under a medium MAT dose. The results were the same for the BMT patients in the same two dimensions (*p* = 0.04 and *p* = 0.03, respectively). In addition, this was the case for the BMT patients only in the next three HCQ dimensions, *namely anticipation of a positive outcome* (*p* = 0.03), *relief from withdrawal or dysphoria* (*p* = 0.04), and *lack of control over use* (*p* = 0.01) on Day 1. These findings show that high MAT doses are accompanied by immediate lower craving levels of both MMT and BMT at the initiation of their dose scheme. The results were different on Day 180 since both the MMT and BMT patients under medium MAT doses exerted lower levels of craving compared to those under high MAT doses. This is indicated by the lower values in the following HCQ dimensions: *desire to use heroin* (*p* = 0.04 and *p* = 0.02 for the MMT and BMT patients, respectively), *intention and planning to use* (*p* = 0.04 for both the MMT and BMT patients), and *lack of control over use* (*p* = 0.01 and *p* = 0.04 for the MMT and BMT patients, respectively). In terms of time, the cravings of both MMT and BMT patients under medium doses were decreased on Day 180 in comparison to Day 1, whereas they remained unaffected for those under high doses. In detail, the score was lower on Day 180 compared to Day 1 for the following HCQ dimensions: *desire to use heroin* (*p* < 0.001 for both the MMT and BMT patients), *intention and planning to use* (*p* < 0.001 for both the MMT and BMT patients), *anticipation of a positive outcome* (*p* = 0.02 and *p* = 0.002 for the MMT and BMT patients, respectively), and *lack of control over use* (*p* = 0.03 and *p* < 0.001 for the MMT and BMT patients, respectively).

### 3.3. Blood Biomarkers

Figure 1 and Figure 2 present the effects of MAT doses on the blood biomarkers of MMT patients. On Day 1, the patients under high doses had lower concentrations of cortisol (*p* = 0.03), EGF (*p* = 0.04), FGF-2 (*p* = 0.01), IL-8 (*p* = 0.02), and IL-10 (*p* < 0.001) than those under medium doses. However, on Day 180, the MMD patients had significantly lower levels of cortisol (*p* = 0.01), FGF-2 (*p* < 0.001), IL-8 (*p* < 0.001), MCP-1 (*p* = 0.005), and TGF-a (*p* = 0.04) compared to their MHD counterparts. No statistically significant effects were observed in EGF, INFγ, IFNa-2, IL-1β, and IL-10. Furthermore, comparing the levels of the measured blood biomarkers of the MMT patients under the same dose at the two examined time points (i.e., Day 1 vs. Day 180), a significant increase in FGF-2 (*p* < 0.001) and IL-8 (*p* < 0.001) in patients under MHD was observed. Moreover, in MMD patients, a significant decrease in cortisol (*p* < 0.001) and MCP-1 (*p* = 0.03) and an increase in IL-8 (*p* < 0.001) levels was observed.

The effects of MAT doses on the blood biomarkers of BMT patients are shown in Figure 3 and Figure 4. On Day 1, the patients under high doses had lower cortisol concentration (*p* < 0.001) than those under medium doses. No statistically significant effects were observed in the other nine biomarkers. On Day 180, the BMD patients had significantly lower levels of cortisol (*p* = 0.03), EGF (*p* = 0.04), FGF-2 (*p* = 0.02), IL-8 (*p* = 0.004), and MCP-1 (*p* = 0.04) compared to their BHD counterparts. No statistically significant effects were observed in INFγ, IFNa-2, IL-1β, IL-10, and TGF-a. Furthermore, comparing the levels of the measured blood biomarkers of the BMT patients under the same dose at the two examined time points (i.e., Day 1 vs. Day 180), a significant increase in IL-8 (*p* < 0.001) was observed on Day 180 compared to Day 1 in the BHD patients. On Day 180, the BMD patients had significantly lower levels of cortisol (*p* < 0.001), EGF (*p* = 0.003), and MCP-1 (*p* < 0.001) and higher levels of IL-8 (*p* < 0.001) compared to Day 1.

### 3.4. Correlation Between Craving and Blood Biomarkers

The results from Pearson’s correlation between the dimensions of HCQ and the pro- and anti-inflammatory biomarkers, as well as growth factors on Day 180, are presented in Table 3 (for the MMT patients) and Table 4 (for the BMT patients). As indicated in Table 3, a significant positive correlation between cortisol and the dimensions of *desire to use heroin* (*p* = 0.032 and *p* = 0.017 for the MHD and MMD patients, respectively) and *intention and planning to use* (*p* = 0.034 and *p* = 0.015 for the MHD and MMD patients, respectively), implying that higher cortisol levels are potentially related to more intense craving. Furthermore, in the MMD patients, negative correlations between EGF (*p* = 0.004), FGF-2 (*p* = 0.013), IFNγ (*p* = 0.004), and IL-8 (*p* = 0.043) and a negative correlation of MCP-1 (*p* = 0.042) in the MHD patients with the dimension *relief from withdrawal or dysphoria* were observed. Additionally, a negative correlation of MCP-1 (*p* = 0.013) with the dimension *anticipation of a positive outcome*, as well as a positive correlation of IL-1β (*p* = 0.018) with *lack of control over use,* and a negative correlation between INFγ (*p* = 0.043) and *lack of control over use* in MMD patients were also found.

Similarly, in the BMT patients (Table 4), significant positive correlations between cortisol (*p* = 0.037 and *p* = 0.041 for the BHD and BMD patients, respectively), IFNγ (*p* = 0.013) in the BMD patients, and IL-8 (*p* = 0.045) and MCP-1 (*p* = 0.002) in the BHD patients with the *desire to use heroin* were found. Moreover, a positive correlation between cortisol (*p* = 0.041 and *p* = 0.045 for the BHD and BMD patients, respectively) and MCP-1 (*p* = 0.005) and *intention and planning to use* in the BMD patients, as well as positive correlations between IL-1β (*p* = 0.045), IL-8 (*p* = 0.011) and MCP-1 (*p* = 0.022) and *anticipation of a positive outcome* in the BHD patients, was also found. Finally, *relief from withdrawal or dysphoria* was negatively correlated with IFNa-2 both in the BMD (*p* = 0.045) and BHD (*p* = 0.037) patients.

## 4. Discussion

The present investigation reports that both the MMT and BMT patients under medium MAT doses had increased craving compared to those under high MAT doses on Day 1. Furthermore, with regards to the time component, the craving of the patients administered with medium methadone and buprenorphine doses was lower than those under high doses on Day 180 in comparison to Day 1. Moreover, the patients under medium MAT doses (both MMT and BMT) exerted decreased inflammation and lower levels of growth factors and cortisol than their high-dose counterparts at the end of the experiment in comparison to Day 1. Finally, crucial correlations between inflammation biomarkers and craving were revealed, implying that inflammatory response may be a factor that leads to high craving levels.

The findings of this study have a crucial clinical significance in terms of the magnitude of MAT doses that clinicians and medical doctors should administer to patients with OUDs. Scarce evidence in the literature indicates that high doses of MAT could induce biochemical or clinical abnormalities. In that direction, a high methadone dose has been related to HPA axis activation, thus contributing to drug-seeking behavior and relapse [47]. Moreover, recent evidence supports that increased MAT doses, especially methadone, may lead to severe clinical complications, such as heart disorders, enhancing dose-dependent stress [48]. In addition, it has been demonstrated that an increased dose of mu opioid agonists (i.e., methadone) is associated with altered methadone tolerance, stress state, craving, and lack of control over use [49], whereas buprenorphine may be carried out for several months without affecting cortisol levels [50]. Furthermore, the results about methadone’s effects on cravings were supported by research findings wherein methadone reduced drug craving at the beginning of the treatment, but thereafter, no progress was noticed [51]. It has to be mentioned that opioid agonists, such as methadone and buprenorphine, suppress acute cortisol levels, even though their levels seem to remain higher in patients with OUDs than in the general population [52]. Notwithstanding, the activation of the HPA axis seems to be further implicated in the immune response by pro-inflammatory agent release.

According to our results, the decreased levels of EGF, FGF-2, and IL-8 on Day 1 were followed by an increase in FGF-2 and IL-8 with a slight enhancement of cortisol levels on Day 180 due to the high methadone dose. Furthermore, in patients under MMD, a significant increase only in IL-8 with parallel reduction of MCP-1 was observed, indicating the immunomodulatory properties of methadone. Similarly, in buprenorphine-treated patients, a significant increase in IL-8 was observed in participants under both doses, namely BHD and BMD, whereas reduced levels of EGF and MCP-1 were found only in BMD patients on Day 180. These findings are consistent with previously reported results wherein prolonged therapy with a mu opioid receptor agonist is potentially related to changes to immune functions, hence leading to stimulation of the immune system, deregulating pro-inflammatory agents, whereas the HPA axis is shifted to a dose-dependent response [16,53,54]. Methadone and buprenorphine, as agonists with high affinity to MORs, bind to their receptors, triggering intracellular signals involving the G-protein receptor kinase (GRK) pathway and β-arrestin 2, resulting in MOR internalization and neurotransmitter modulation release. This happens through the receptor tyrosine kinase (RTK) pathway [55]. Thus, a cross-talk mechanism between MORs and the epidermal growth factor receptor (EGFR) and fibroblast growth factor receptor (FGFR) through their activation by EGF and FGF-2, respectively, is involved in the RTK pathway inducing side effects [55,56,57]. It seems that the effect of both EGF and FGF-2 on glutamatergic and dopaminergic neuron activity, which is strongly implicated in OUDs, plays a substantial role in dopamine balance in the reward system [58,59]. It has been reported that increased FGF-2 levels are related to psychiatric disorders and may act as a predisposing factor for anxiety and as a modulator of environmental influences on stress behavior, whereas elevated levels of EGF are implicated in psychotic symptoms through deregulation of dopaminergic signaling [60,61,62].

In addition, the significant decrease in MCP-1 with a parallel increase in IL-8 levels in medium methadone and buprenorphine doses that were found herein might reinforce the idea that an underlying inflammatory process has occurred in patients under MAT. MCP-1, as a chemokine, is produced by many cell types, including astrocytic and microglial cells, after induction of oxidative stress and cytokines or growth factors release, inducing monocytes chemotaxis through the activation of G-protein-coupled receptors [43]. Findings have suggested that MCP-1 may play a crucial role in drug addiction, participating in dopamine neuron activation at the reward system and affecting HPA axis deregulation through the secretion of corticotrophin-releasing factor [63,64,65]. The results herein are supported by the literature as buprenorphine may reduce MCP-1 inhibiting monocytes by blocking the integrins pathway [28] and by the action of buprenorphine in addiction control and neuroinflammatory process deterioration [29]. Furthermore, concerning IL-8, the results in the present investigation are consistent with previous findings wherein higher levels of this cytokine were observed in patients under methadone, indicating that it more severely affects the intracellular functions than buprenorphine [11,66,67]. Although the mechanism between opioids and cytokine production in the central nervous system is still not clear, cytokine expression in OUD patients has been detected in glial cells [12]. Furthermore, the presence of opioid receptors in astrocytes suggests a potential interaction between opioids and glial cells and also neuroinflammation along with hyperactivation of the HPA axis due to prolonged opioid administration [68,69,70,71]. One plausible mechanism may involve the ΔFosB transcription factor and its role in chronic exposure to drugs of abuse and addiction behavior in the reward system and in the paraventricular nucleus of the hypothalamus, hence in the HPA axis [72,73]. Research findings have shown that glucocorticoids may potentially regulate ΔFosB expression after chronic opioid administration as an underlying mechanism that links stress hormones, like cortisol, with opioid addiction [74]. Furthermore, ΔFosB, in long-term opioid exposure, targets the nuclear factor kappa-light-chain-enhancer of activated B cells (NF-κB), which is also stimulated by Toll-like receptor-4 (TRL-4), leading to inflammatory response through cytokine release [75,76]. This mechanism seems to be dose-dependent since animal studies have demonstrated a relationship between ΔFosB expression and dose of addictive substances, wherein higher doses exert ΔFosB overexpression [77]. In addition, oxidative stress might be a mediator in the above interaction since recent findings revealed that both methadone and buprenorphine induce oxidative stress, although at a lower level than other illicit opioids [5]. This finding implies that increased production of pro-inflammatory agents induces oxidative imbalance that further promotes inflammation [78].

Moreover, comparing the different doses in both medications tested herein, the opioid-related inflammatory process seems to be dose-dependent since a higher dose exerts more severe effects on the function of the immune system, considering the elevated levels of pro-inflammatory biomarkers on Day 180. Scarce studies on humans have shown that increasing methadone dose appears to be highly related to brain white-matter integrity disturbance, affecting dopamine transporters circulation and cytokine production and potentially having a detrimental impact on brain function through enhanced inflammatory process [11,79,80]. Regarding buprenorphine, it is initially metabolized in norbuprenorphine, which is featured as a full agonist on MORs and a partial agonist on KORs [81]. According to the literature, norbuprenorphine could compete with buprenorphine in binding with the receptors, appearing to have 10 times more of a depressant effect on the respiratory system than buprenorphine, along with the tolerance in high doses—a fact that is not observed at lower doses [82,83,84].

Based on our findings through correlation, a direct significant relation of EGF, FGF-2, IL-8, IL-10, MCP-1, and TGF-a with the HCQ dimensions on both medication doses was revealed on Day 180. This result depicts that methadone and buprenorphine affect and even dysregulate the normal function of the immune system. Previous studies have shown that TNF-a, TGF-a, IL-8, and IL-5 were increased in OUD patients, whilst others have reported stable or decreased pro-inflammatory responses [85,86]. Moreover, taken together the correlation of cytokines and cortisol with the HCQ dimensions in both medications, the hypothesis that MAT may modulate HPA axis function affecting immune response must be under strong consideration. Indeed, patients under long-term MAT suffer from physical and psychological chronic stress that causes HPA axis activation with potentially detrimental impact on their immunity and changes in behavior, metabolism, and autonomic function [51,87].

The present investigation also has specific limitations. First, the majority of the patients were male since the number of females who attend MAT programs is much smaller, thus leading to a lack of generalizability, as is the case with all interventional studies [88]. Secondly, because MAT patients are vulnerable to diverse exogenous stressors, it is difficult to strictly determine whether the immune–endocrine changes have been induced by medications or by the patient’s lifestyle since lifestyle and activities of daily living, as stress, sleep, or alcohol consumption are known to disturb cell function [10]. Research findings imply that there is an underpinned relation between drug dependence and environmental factors interacting with psycho-endocrine systems, as they appear to be more sensitive to social disruptions with a dysregulation of the hippocampal glucocorticoid receptors [89,90]. Furthermore, the interaction between socio-economic level and DNA alterations that has been demonstrated indicates the impact of environmental stimuli on the physiological and behavioral responses related to drug-seeking regimens [89]. It has been shown that craving and drug-seeking behavior triggered by lifestyle activities or social factors, such as social isolation, induce neurogenic changes in the function of dopaminergic and serotonergic systems that potentially lead to modified neurotransmission, affecting the retention of treatment and increasing the risk of relapse [91,92]. In addition, no correlation was observed between immuno-endocrine measurements and psychiatric comorbidity, considering that psychiatric diseases are accompanied by HPA axis activation and cytokines release [93]. According to the literature, substance use is a crucial factor in the severity of mental disorders. Indeed, the development of opioid-administration behaviors and the deregulation of the stress-response system on the HPA axis have been associated with poor treatment adherence, longer and more frequent mood disorders, enhanced frequency of depressive episodes, and lower functional recovery [94,95]. The presence of higher peak cortisol in response to induced stressors/cues in substance abuse dropouts found by Daughters et al. [96] and Fatseas et al. [31] indicated that a distorted HPA axis may exist in substance abusers. Finally, it has to be mentioned that HCQ is a self-reported instrument; hence, a risk of bias is always possible.

## 5. Conclusions

In conclusion, the present study shows that high doses of methadone and buprenorphine negatively influence the normal function of the immune system in MAT patients. Furthermore, they appear to be able to dysregulate normal cellular functions in terms of dose magnitude due to the alterations in immune–endocrine agents reported. Of note, methadone and buprenorphine are key MAT approaches, and high doses of both medications are effective for the initial treatment and stabilization process. Nevertheless, they may induce side effects after chronic administration, and often, dose modifications may putatively lead to relapse. Therefore, it is suggested that clinicians should reconsider the necessity of maintaining MAT doses for long periods of time. Further research on the pharmacokinetic and pharmacodynamic properties of both medications is needed, along with in vivo human data, especially for high doses, to better apply MAT schemes to patients with OUDs.

## Figures and Tables

**Figure 1 pharmacy-13-00040-f001:**
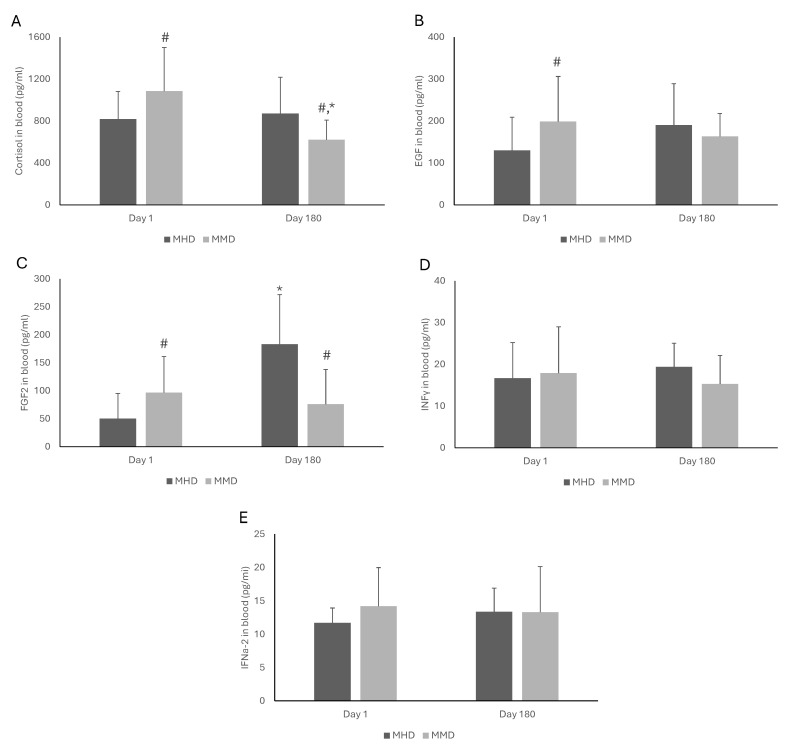
The effects of methadone medium dose (MMD) and methadone high dose (MHD) on the concentrations of cortisol (**A**), EGF (**B**), FGF2 (**C**), IFNγ (**D**), and IFNa-2 (**E**) in the blood of patients under medication for addiction treatment. The statistical analysis of the results was performed through two-way ANOVA with repeated measures. *: Statistically significant result comparing Days 1 and 180 in the same dose. #: Statistically significant result comparing MMD and MHD at the same time point. EGF: basic epidermal growth factor; FGF-2: fibroblast growth factor-2; INFγ: interferon gamma; IFNa-2: interferon alpha-2.

**Figure 2 pharmacy-13-00040-f002:**
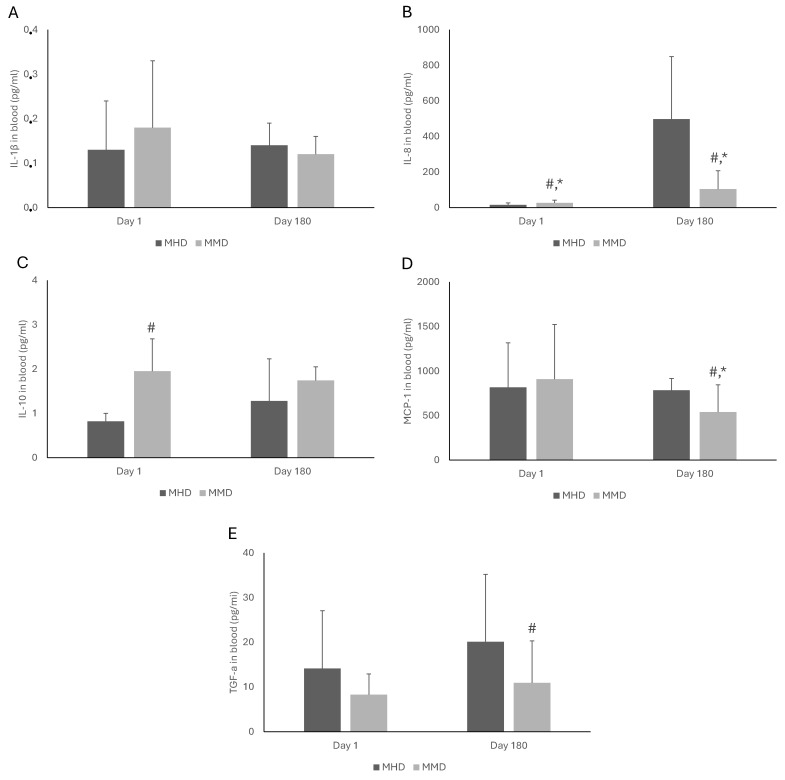
The effects of methadone medium dose (MMD) and methadone high dose (MHD) on the concentrations of IL-1β (**A**), IL-8 (**B**), IL-10 (**C**), MCP-1 (**D**), and TGF-a (**E**) in the blood of patients under medication for addiction treatment. The statistical analysis of the results was performed through two-way ANOVA with repeated measures. *: Statistically significant result comparing Days 1 and 180 in the same dose. #: Statistically significant result comparing MMD and MHD at the same time point. IL-1β: interleukin-1 beta; IL-8: interleukin-8; IL-10: interleukin-10; MCP-1: monocyte chemoattractant protein-1; TGF-a: transforming growth factor-alpha.

**Figure 3 pharmacy-13-00040-f003:**
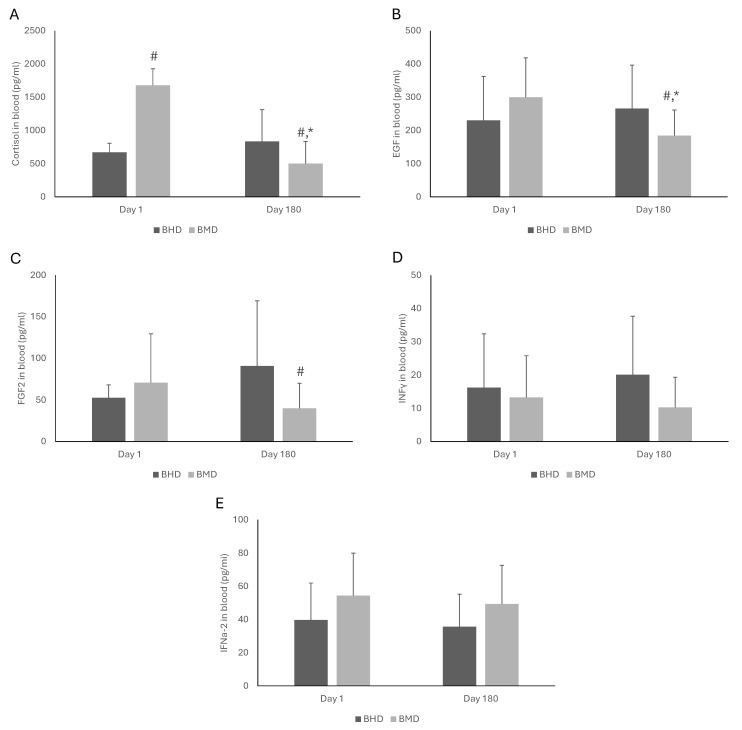
The effects of buprenorphine medium dose (BMD) and buprenorphine high dose (BHD) on the concentrations of cortisol (**A**), EGF (**B**), FGF2 (**C**), IFNγ (**D**), and IFNa-2 (**E**) in the blood of patients under medication for addiction treatment. The statistical analysis of the results was performed through two-way ANOVA with repeated measures. *: Statistically significant result comparing Days 1 and 180 in the same dose. #: Statistically significant result comparing MMD and MHD at the same time point. EGF: basic epidermal growth factor; FGF-2: fibroblast growth factor-2; INFγ: interferon gamma; IFNa-2: interferon alpha-2.

**Figure 4 pharmacy-13-00040-f004:**
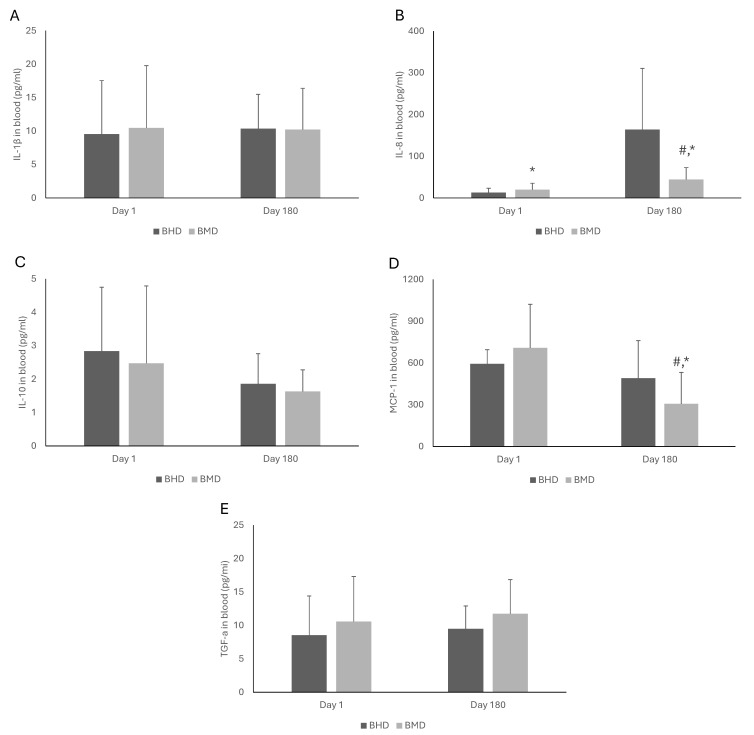
The effects of buprenorphine medium dose (BMD) and buprenorphine high dose (BHD) on the concentrations of IL-1β (**A**), IL-8 (**B**), IL-10 (**C**), MCP-1 (**D**), and TGF-a (**E**) in the blood of patients under medication for addiction treatment. The statistical analysis of the results was performed through two-way ANOVA with repeated measures. *: Statistically significant result comparing Days 1 and 180 in the same dose. #: Statistically significant result comparing MMD and MHD at the same time point. IL-1β: interleukin-1 beta; IL-8: interleukin-8; IL-10: interleukin-10; MCP-1: monocyte chemoattractant protein-1; TGF-a: transforming growth factor-alpha.

**Table 1 pharmacy-13-00040-t001:** The demographic characteristics of the participants as a whole and per group.

Characteristic		Groups			
	Total (n = 66)	MHD	BHD	MMD	BMD
Sex, n (%)					
*Males*	47 (71.2)	11 (64.7)	11 (64.7)	12 (70.58)	13 (86.6)
*Females*	19 (28.8)	6 (35.3)	6 (35.3)	5 (29.42)	2 (13.4)
Age (years), n (%)					
*18–24*	4 (6.1)	0 (0.0)	2 (11.8)	2 (11.8)	0 (0.0)
*25–30*	9 (13.6)	2 (11.8)	3 (17.6)	2 (11.7)	2 (13.3)
*31–40*	23 (34.8)	7 (41.2)	4 (23.5)	6 (35.3)	6 (40.0)
*41–50*	25 (37.9)	6 (35.2)	7 (41.2)	6 (35.3)	6 (40.0)
*51–60*	5 (7.6)	2 (11.8)	1 (5.9)	1 (5.9)	1 (6.7)
Educational status, n (%)					
*Primary school*	16 (24.2)	6 (35.3)	3 (17.6)	4 (23.5)	3 (20.0)
*Middle/high school*	38 (57.6)	8 (47.1)	8 (47.1)	10 (58.9)	12 (80.0)
*University/postgraduate*	12 (18.2)	3 (17.6)	6 (35.3)	3 (17.6)	0 (0.0)
Family status, n (%)					
*Married*	12 (18.1)	4 (23.5)	3 (17.6)	2 (11.8)	3 (20.0)
*Unmarried*	35 (53.0)	9 (53.0)	8 (47.1)	10 (58.8)	8 (53.4)
*Widowed*	4 (6.1)	3 (17.6)	1 (5.9)	0 (0.0)	0 (0.0)
*Divorced*	11 (16.7)	1 (5.9)	4 (23.5)	2 (11.8)	4 (26.6)
*Separated*	4 (6.1)	0 (0.0)	1 (5.9)	3 (17.6)	0 (0.0)
Place of residence, n (%)					
*Urban*	50 (75.8)	15 (88.2)	13 (76.5)	11 (64.7)	11 (73.3)
*Rural*	16 (24.2)	2 (11.8)	4 (23.5)	6 (35.3)	4 (26.7)
Age at first use, mean (SD)	18.4 (5.1)	17.6 (4.7)	18.5 (4.1)	17.5 (4.5)	20.3 (6.4)
Years of substance abuse, mean (SD)	16.9 (8.5)	15.4 (9.3)	20.4 (8.6)	14.1 (6.8)	19.6 (7.9)

SD: standard deviation; MHD: methadone high dose; BHD: buprenorphine high dose; MMD: methadone medium dose; BMD: buprenorphine medium dose.

**Table 2 pharmacy-13-00040-t002:** The effects of MAT doses on craving through dimensions of heroin craving questionnaire.

	Day 1				Day 180							
HCQ Dimensions	MHD	MMD	BHD	BMD	MHD	MMD	BHD	BMD	MHD 1 vs. 180	MMD 1 vs. 180	BHD 1 vs. 180	BMD 1 vs. 180
	Mean Score ± SD								*p*			
D.U.	21.7 ± 9.7	27.8 ± 7.6	20.3 ± 5.7	25.8 ± 9.3	20.6 ± 6.4	16.3 ± 5.6	19.2 ± 5.3	15.3 ± 4.1	0.69	**<0.001**	0.56	**<0.001**
*p*	**0.04**		**0.04**		**0.04**		**0.02**					
I.P.U.	22.9 ± 7.1	27.5 ± 5.8	20.1 ± 5.8	25.6 ± 8.1	22.1 ± 7.3	17.6 ± 5.2	18.8 ± 8.5	14.1 ± 4.1	0.74	**<0.001**	0.60	**<0.001**
*p*	**0.04**		**0.03**		**0.04**		**0.04**					
A.P.O.	21.4 ± 10.5	25.7 ± 11.9	19.6 ± 9.1	25.9 ± 7.2	22.9 ± 12.6	16.4 ± 11.2	18.5 ± 12.5	15.4 ± 9.9	0.70	**0.02**	0.77	**0.002**
*p*	0.27		**0.03**		0.12		0.40					
R.W.D.	29.0 ± 13.0	29.8 ± 12.2	24.9 ± 7.2	29.9 ± 5.8	27.9 ± 9.3	24.5 ± 11.5	24.8 ± 12	25.5 ± 10.9	0.77	0.20	0.97	0.17
*p*	0.85		**0.04**		0.35		0.86					
L.C.U.	26.6 ± 9.9	26.9 ± 8.6	22.8 ± 7.1	30.5 ± 9.6	27.8 ± 8.7	21.0 ± 6.7	24.4 ± 7.9	19.1 ± 6.1	0.70	**0.03**	0.53	**<0.001**
*p*	0.92		**0.01**		**0.01**		**0.04**					

SD: standard deviation; MAT: medication for addiction treatment; MHD: methadone high dose; MMD: methadone medium dose; BHD: buprenorphine high dose; BMD: buprenorphine medium dose; D.U: desire to use heroin; I.P.U.: intention and planning to use; A.P.O.: anticipation of a positive outcome; R.W.D.: relief from withdrawal or dysphoria; L.C.U.: lack of control over use. Bold numbers indicate statistically significant results (*p* < 0.05).

**Table 3 pharmacy-13-00040-t003:** Pearson’s correlation results between the HCQ dimensions and pro-/anti-inflammatory biomarkers as well as growth factors of MHD and MMD patients on Day 180.

Biomarkers		Desire to Use Heroin		Intention and Planning to Use		Anticipation of Positive Outcome		Relief from Withdrawal or Dysphoria		Lack of Control over Use	
		MHD	MMD	MHD	MMD	MHD	MMD	MHD	MMD	MHD	MMD
Cortisol	r	0.75	0.50	0.75	0.52	−0.47	−0.25	−0.23	−0.15	−0.13	−0.18
	*p*	**0.032**	**0.017**	**0.034**	**0.015**	0.207	0.303	0.626	0.527	0.742	0.457
EGF	r	0.03	0.01	0.31	0.03	0.55	0.27	0.36	−0.57	0.32	−0.11
	*p*	0.948	0.983	0.458	0.904	0.127	0.256	0.422	**0.004**	0.396	0.648
FGF-2	r	−0.49	0.24	−0.27	0.20	−0.01	0.13	0.25	−0.56	−0.12	−0.02
	*p*	0.215	0.298	0.511	0.397	0.982	0.591	0.596	**0.013**	0.751	0.934
IFNγ	r	−0.05	0.49	0.05	0.27	−0.16	0.17	−0.23	−0.62	−0.70	0.04
	*p*	0.914	**0.028**	0.900	0.257	0.673	0.484	0.622	**0.004**	**0.043**	0.854
IFNa-2	r	−0.80	−0.07	−0.45	−0.15	−0.08	0.12	0.17	0.17	−0.05	−0.23
	*p*	**0.017**	0.755	0.263	0.526	0.841	0.639	0.709	0.495	0.890	0.333
IL-1β	r	−0.02	−0.05	−0.03	−0.08	−0.12	0.08	−0.13	0.11	−0.09	0.53
	*p*	0.962	0.835	0.951	0.750	0.760	0.736	0.780	0.645	0.812	**0.018**
IL-8	r	0.02	0.25	−0.01	0.29	0.12	0.31	−0.68	−0.47	−0.12	−0.25
	*p*	0.961	0.280	0.989	0.215	0.750	0.203	0.093	**0.043**	0.751	0.297
IL-10	r	−0.01	0.20	−0.04	0.15	0.10	0.08	−0.68	0.10	−0.16	−0.04
	*p*	0.985	0.397	0.926	0.536	0.800	0.734	0.094	0.694	0.674	0.862
MCP-1	r	−0.37	0.18	−0.29	0.27	−0.09	−0.56	−0.71	−0.16	0.06	−0.14
	*p*	0.367	0.435	0.486	0.246	0.827	**0.013**	**0.042**	0.511	0.882	0.556
TGF-a	r	−0.21	−0.14	−0.18	−0.22	0.10	−0.16	−0.35	−0.11	0.06	0.02
	*p*	0.620	0.746	0.673	0.600	0.800	0.767	0.446	0.801	0.887	0.969

MHD: methadone high dose; MMD: methadone medium dose; r: Pearson’s correlation coefficient; EGF: basic epidermal growth factor; FGF-2: fibroblast growth factor-2; INFγ: interferon gamma; IFNa-2: interferon alpha-2; IL-1β: interleukin-1 beta; IL-8: interleukin-8; IL-10: interleukin-10; MCP-1: monocyte chemoattractant protein-1; TGF-a: transforming growth factor-alpha. Bold numbers indicate statistical significance (*p* < 0.05).

**Table 4 pharmacy-13-00040-t004:** Pearson’s correlation results between the HCQ dimensions and pro-/anti-inflammatory biomarkers as well as growth factors of BHD and BMD patients on Day 180.

Biomarkers		Desire to Use Heroin		Intention and Planning to Use		Anticipation of Positive Outcome		Relief from Withdrawal or Dysphoria		Lack of Control over Use	
		BHD	BMD	BHD	BMD	BHD	BMD	BHD	BMD	BHD	BMD
Cortisol	r	0.61	0.74	0.66	0.76	−0.29	−0.05	0.29	−0.07	0.01	0.04
	*p*	**0.037**	**0.041**	**0.041**	**0.045**	0.380	0.918	0.415	0.869	0.981	0.930
EGF	r	−0.14	−0.16	−0.04	0.911	−0.05	−0.01	0.36	0.764	0.09	0.656
	*p*	0.681	0.699	0.918	−0.02	0.873	0.758	0.305	−0.42	0.779	0.15
FGF-2	r	−0.05	−0.19	−0.21	0.955	−0.19	−0.13	0.08	0.305	0.15	0.731
	*p*	0.882	0.656	0.533	−0.14	0.568	0.983	0.828	−0.29	0.632	−0.05
IFNγ	r	−0.22	0.82	−0.15	0.748	−0.22	0.91	0.01	0.491	−0.26	0.910
	*p*	0.508	**0.013**	0.668	0.53	0.520	0.804	0.984	0.69	0.420	0.57
IFNa-2	r	−0.18	−0.29	0.02	0.174	0.03	−0.34	−0.65	−0.20	0.15	0.143
	*p*	0.588	0.492	0.955	−0.38	0.926	0.012	**0.037**	**0.049**	0.652	−0.16
IL-1β	r	−0.12	−0.30	0.06	0.30	−0.62	0.17	0.14	0.19	−0.13	0.18
	*p*	0.729	0.465	0.869	0.466	**0.045**	0.752	0.708	0.652	0.691	0.674
IL-8	r	0.60	0.55	−0.35	0.10	−0.68	0.57	−0.13	0.63	−0.22	0.06
	*p*	**0.045**	0.158	0.297	0.822	**0.011**	0.235	0.711	0.094	0.495	0.890
IL-10	r	0.18	0.41	0.25	0.29	−0.02	0.35	−0.53	0.61	0.09	0.45
	*p*	0.596	0.204	0.459	0.488	0.952	0.464	0.114	0.105	0.770	0.262
MCP-1	r	0.82	0.11	−0.57	0.89	−0.68	0.28	−0.44	−0.03	−0.47	−0.18
	*p*	**0.002**	0.793	0.070	**0.005**	**0.022**	0.587	0.201	0.948	0.119	0.667
TGF-a	r	−0.17	0.12	−0.02	0.25	0.02	0.21	0.07	0.14	−0.23	0.11
	*p*	0.620	0.610	0.945	0.296	0.953	0.386	0.840	0.555	0.463	0.632

BHD: buprenorphine high dose; BMD: buprenorphine medium dose; r: Pearson’s correlation coefficient; EGF: basic epidermal growth factor; FGF-2: fibroblast growth factor-2; INFγ: interferon gamma; IFNa-2: interferon alpha-2; IL-1β: interleukin-1 beta; IL-8: interleukin-8; IL-10: interleukin-10; MCP-1: monocyte chemoattractant protein-1; TGF-a: transforming growth factor-alpha. Bold numbers indicate statistical significance (*p* < 0.05).

## Data Availability

The data used to support the findings of this study are available upon request.

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
