# Peer review of "Methadone and Buprenorphine as Medication for Addiction Treatment Diversely Affect Inflammation and Craving Depending on Their Doses"

_pharmacy, 2025, doi:10.3390/pharmacy13020040_

Round 1

Reviewer 1 Report

Comments and Suggestions for Authors

The authors investigated the dose-dependent effects of methadone and buprenorphine on craving, inflammatory biomarkers, and cortisol levels in patients undergoing medication-assisted treatment for opioid use disorders. To the reviewer’s point of view, this manuscript makes an important contribution to the understanding of dose-dependent effects of methadone and buprenorphine in MAT programs. However, the manuscript would benefit from further clarifications and a more comprehensive discussion, particularly regarding its limitations. Addressing these aspects would substantially enhance the manuscript's scientific rigor, impact, and overall relevance.

  • Why did the authors specifically select the inflammation markers? The manuscript does not explain why these specific markers were prioritized over others (CRP, etc.);

·         The authors should provide a more detailed explanation of the observed differences between high and medium doses, particularly regarding the dose-dependent effects on biomarkers like IL-8 and cortisol;

·         The manuscript acknowledges lifestyle factors and activities of daily living as limitations but does not provide a detailed discussion on how these variables might influence the generalizability of the results, particularly in the context of their potential impact on immune and endocrine responses;

·         The authors could elaborate on the role of psychiatric comorbidities in influencing HPA axis activation and their potential interplay with immune and endocrine responses, as these conditions are often prevalent in patients with opioid use disorders and may significantly affect the outcomes of medication-assisted treatment;

·         The proposed mechanisms linking MAT-induced biomarker changes to craving reduction could be explored in greater depth, with a detailed discussion of the molecular pathways involved, such as the interaction between opioid receptors, the HPA axis, and pro-inflammatory cytokines. Additionally, elaborating on the clinical implications of these pathways could provide valuable insights into optimizing treatment strategies for opioid use disorder.

Author Response

Comment 1: The authors investigated the dose-dependent effects of methadone and buprenorphine on craving, inflammatory biomarkers, and cortisol levels in patients undergoing medication-assisted treatment for opioid use disorders. To the reviewer’s point of view, this manuscript makes an important contribution to the understanding of dose-dependent effects of methadone and buprenorphine in MAT programs.

Response: We would like to thank Reviewer 1 for his/her kind comment.

Comment 2: However, the manuscript would benefit from further clarifications and a more comprehensive discussion, particularly regarding its limitations. Addressing these aspects would substantially enhance the manuscript's scientific rigor, impact, and overall relevance.

Response: According to Reviewer’s request, we have discussed the limitations by adding the appropriate text in “Discussion”. The added text is presented in Comments 5 and 6.

Comment 3: Why did the authors specifically select the inflammation markers? The manuscript does not explain why these specific markers were prioritized over others (CRP, etc.);

Response: We thank Reviewer 1 for giving us the opportunity to clear this up. Based on the literature, opioid craving as a chronic stressful condition, potentially may activate inflammatory response releasing inflammatory substances by central and peripheral biomarkers including hormones (cortisol), growth factors, and cytokines (Davis et al. 2023). Regarding C-reactive protein (CRP), it can be elevated during the acute phase of an inflammatory process as a non-specific inflammatory biomarker, since according to research findings, both medication for addiction treatment, i.e., methadone and buprenorphine, could maintain CRP in low levels (Lu et al. 2019).

Davis, S. L., Latimer, M., & Rice, M. (2023). Biomarkers of Stress and Inflammation in Children. Biological research for nursing, 25(4), 559–570. https://doi.org/10.1177/10998004231168805.

Lu, R. B., Wang, T. Y., Lee, S. Y., Chen, S. L., Chang, Y. H., See Chen, P., Lin, S. H., Chu, C. H., Huang, S. Y., Tzeng, N. S., Lee, I. H., Chin Chen, K., Kuang Yang, Y., Chen, P., Chen, S. H., & Hong, J. S. (2019). Correlation between interleukin-6 levels and methadone maintenance therapy outcomes. Drug and alcohol dependence, 204, 107516. https://doi.org/10.1016/j.drugalcdep.2019.06.018.   

Comment 4: The authors should provide a more detailed explanation of the observed differences between high and medium doses, particularly regarding the dose-dependent effects on biomarkers like IL-8 and cortisol;

Response: According to Reviewer’s request, even though scarce evidence exists in literature, however we have made an effort to explain further the interrelationship between HPA axis stimulation and proinflammatory cytokine release by adding the following text in the “Discussion”: “One plausible mechanism may involve ΔFosB transcription factor, and its role in chronic exposure to drugs of abuse and addiction behavior in the reward system, and in the paraventricular nucleus of the hypothalamus, hence, in the HPA axis [72,73]. Research findings have shown that glucocorticoids may potentially regulate ΔFosB expression after chronic opioid administration, as an underlying mechanism that links stress hormones like cortisol, with opioid addiction [74]. Furthermore, ΔFosB, in long-term opioid exposure, targets to nuclear factor kappa-light-chain-enhancer of activated B cells (NF-κB) which is also stimulated by Toll like receptor-4 (TRL-4), leading to inflammatory response through cytokine release [75,76]. This mechanism seems to be dose-dependent, since animal studies have demonstrated a relationship between ΔFosB expression and dose of addictive substances, wherein higher doses exert ΔFosB overexpression [77].

Comment 5: The manuscript acknowledges lifestyle factors and activities of daily living as limitations but does not provide a detailed discussion on how these variables might influence the generalizability of the results, particularly in the context of their potential impact on immune and endocrine responses;

Response: Long-term opioid drug use, as a chronic stressful condition, is mediated through reward system leading to the activation of biological factors (i.e., corticotrophin-releasing factor, cortisol, pro-inflammatory agents), which promote the development of negative emotions increasing the risk for relapse and overdose. Based on Reviewer’s comment, we have added the following text in the “Discussion”: Research findings imply that there is an underpinned relation between drug dependence and environmental factors interacting with psycho-endocrine systems, as they appear to be more sensitive to social disruptions with a dysregulation of the hippocampal glucocorticoid receptors [89,90]. Furthermore, the interaction between socio-economic level and DNA-alterations that has been demonstrated indicates the impact of environmental stimuli on the physiological and behavioral responses related to drug-seeking regimen [89]. It has been shown that craving and drug-seeking behavior triggered by life-style activities or social factors, as social isolation, induce neurogenic changes in the function of domanimergic and serotonergic systems that potentially lead to modified neurotransmission affecting the retention to treatment, and increasing the risk for relapse [91,92]”.

Comment 6: The authors could elaborate on the role of psychiatric comorbidities in influencing HPA axis activation and their potential interplay with immune and endocrine responses, as these conditions are often prevalent in patients with opioid use disorders and may significantly affect the outcomes of medication-assisted treatment;

Response: According to this comment, the following text has been added in “Discussion”: “According to the literature, substance use is a crucial factor for the severity of mental disorders. Indeed, the development of opioid-administration behaviors and the deregulation of stress-responsive system on HPA axis have been associated with poor treatment adherence, longer, and frequent mood disorders, enhanced frequency of depressive episodes, and lower functional recovery [94,95]”.

Comment 7: The proposed mechanisms linking MAT-induced biomarker changes to craving reduction could be explored in greater depth, with a detailed discussion of the molecular pathways involved, such as the interaction between opioid receptors, the HPA axis, and pro-inflammatory cytokines. Additionally, elaborating on the clinical implications of these pathways could provide valuable insights into optimizing treatment strategies for opioid use disorder.

Response: As we have written in our response for Comment 4, we have added a text in “Discussion” in order to describe a possible mechanism cross-talking between opioid receptors, HPA axis and proinflammatory agents. Furthermore, in “Conclusion”, a text about further strategies has already been written: Therefore, it is suggested that clinicians should reconsider the necessity of maintaining MAT doses for long periods of time”.

Reviewer 2 Report

Comments and Suggestions for Authors

This study investigated the effects of buprenorphine and methadone, commonly used in medication-assisted treatment (MAT) for opioid use disorder, on craving, inflammation biomarkers, and cortisol levels, with a focus on dosage. Sixty-six patients (34 on methadone and 32 on buprenorphine) were stabilized on either high or medium doses of these medications and monitored over 180 days.

Key findings include:

1) High doses of both medications reduced craving, cortisol, and inflammation at Day 1, but these effects were not sustained by Day 180.

2) Medium doses showed consistent reductions in craving and biochemical markers over time, indicating sustained positive effects.

3) Adjusting MAT dosages shortly after stabilization is recommended to prevent inflammation, reduce relapse risk, and support long-term rehabilitation for opioid-addicted patients. This highlights the importance of personalized dosage adjustments in MAT programs.

Comments

The manuscript is interesting and presents new and useful contexts. There are no publications in the literature on this subject.

Title: correctly describes the contents of the manuscript.

Abstract: provides a clear and concise overview of the content and conclusions of the paper without having to read the entire text.

Introduction: provides sufficient background.

Tables: provide detailed information although they are not easy to understand for a reader. A graphical representation of the Tables, could help with faster interpretation of the results. It is requested to supplement Table values with bar diagrams.

Conclusions: are consistent with the evidence and the limitations of this study are already considered by the authors.

References: are appropriate and exhaustive.

Author Response

Comment 1: The manuscript is interesting and presents new and useful contexts. There are no publications in the literature on this subject.

Response: We would like to thank Reviewer 2 for his/her kind comment.

Comment 2: Title: correctly describes the contents of the manuscript.

Response: Thank you again.

Comment 3: Abstract: provides a clear and concise overview of the content and conclusions of the paper without having to read the entire text.

Response: Thank you again.

Comment 4: Introduction: provides sufficient background.

Response: Thank you for the kind comment.

Comment 5: Tables: provide detailed information although they are not easy to understand for a reader. A graphical representation of the Tables, could help with faster interpretation of the results. It is requested to supplement Table values with bar diagrams.

Response: We agree with Reviewer 2 that the Tables may confuse the readership to understand the results of our study. Therefore, we have replaced in the revised manuscript Tables 3 and 4 with Figures 1, 2, 3 and 4.

Comment 6: Conclusions: are consistent with the evidence and the limitations of this study are already considered by the authors.

Response: Thank you for the kind comment.

Comment 7: References: are appropriate and exhaustive.

Response: Thank you for the kind comment.

Reviewer 3 Report

Comments and Suggestions for Authors

The results of this sutdy are very complicated.  As the comments of the authors in 'Discussion', the investigation of this manuscript has limitations. Many readers are probably difficult to understand what the authors newly found out in this study only by using the results on Tables.  In addition, the authors should emphasize meaningful value shifts of biomarkers on all Tables. They need clear comments about some reasons for the shifts. Futhermore, this reviewer think in a question what the various shifts cause to the patients. 

Author Response

Comment 1: The results of this study are very complicated. As the comments of the authors in 'Discussion', the investigation of this manuscript has limitations. Many readers are probably difficult to understand what the authors newly found out in this study only by using the results on Tables. In addition, the authors should emphasize meaningful value shifts of biomarkers on all Tables. They need clear comments about some reasons for the shifts. Furthermore, this reviewer think in a question what the various shifts cause to the patients.

Response: We would like to inform Reviewer 3 that we have done our best to improve our manuscript and make our results clearer to the readership. Therefore, we have replaced in the revised manuscript Tables 3 and 4 with Figures 1, 2, 3 and 4. Regarding limitations of the study, we have added the following text in the “Discussion”: “Research findings imply that there is an underpinned relation between drug dependence and environmental factors interacting with psycho-endocrine systems, as they appear to be more sensitive to social disruptions with a dysregulation of the hippocampal glucocorticoid receptors [89,90]. Furthermore, the interaction between socio-economic level and DNA-alterations that has been demonstrated indicates the impact of environmental stimuli on the physiological and behavioral responses related to drug-seeking regimen [89]. It has been shown that craving and drug-seeking behavior triggered by life-style activities or social factors, as social isolation, induce neurogenic changes in the function of domanimergic and serotonergic systems that potentially lead to modified neurotransmission affecting the retention to treatment, and increasing the risk for relapse [91,92]”.

Concerning the biological plausibility of our results, we have added the following text in “Discussion”: “One plausible mechanism may involve ΔFosB transcription factor, and its role in chronic exposure to drugs of abuse and addiction behavior in the reward system, and in the paraventricular nucleus of the hypothalamus, hence, in the HPA axis [72,73]. Research findings have shown that glucocorticoids may potentially regulate ΔFosB expression after chronic opioid administration, as an underlying mechanism that links stress hormones like cortisol, with opioid addiction [74]. Furthermore, ΔFosB, in long-term opioid exposure, targets to nuclear factor kappa-light-chain-enhancer of activated B cells (NF-κB) which is also stimulated by Toll like receptor-4 (TRL-4), leading to inflammatory response through cytokine release [75,76]. This mechanism seems to be dose-dependent, since animal studies have demonstrated a relationship between ΔFosB expression and dose of addictive substances, wherein higher doses exert ΔFosB overexpression [77]”; and “According to the literature, substance use is a crucial factor for the severity of mental disorders. Indeed, the development of opioid-administration behaviors and the deregulation of stress-responsive system on HPA axis have been associated with poor treatment adherence, longer, and frequent mood disorders, enhanced frequency of depressive episodes, and lower functional recovery [94,95]”.

Round 2

Reviewer 2 Report

Comments and Suggestions for Authors

The manuscript was amended as requested so it can now be accepted in present form.

Reviewer 3 Report

Comments and Suggestions for Authors

The authors carefully revised their original manuscript based on the reviewers’ comments. In the revised manuscript, they described suitable comments as their answers to the reviewers’ questions. This reviewer recommends this manuscript as an acceptable article in “Pharmacy”, after the authors respond to the following points.   

1)     The authors need to add the details of statistical tests in Figures 1–4 to their legends.  In addition, they must describe the procedure of their statistical analysis of this study to the experimental section of the revised manuscript.

Author Response

Response to Reviewer Comments (round 2)

1. Summary

Thank you very much for taking the time to review this manuscript. Please find the detailed responses below and the corresponding revisions/corrections highlighted/in track changes in the re-resubmitted files.

2. Questions for General Evaluation

Reviewer’s Evaluation

Response and Revisions

Does the introduction provide sufficient background and include all relevant references?

Yes

Are all the cited references relevant to the research?

Yes

Is the research design appropriate?

Yes

Are the methods adequately described?

Must be improved

According to Reviewer’s suggestions, revisions have been made, as indicated in following report. 

Are the results clearly presented?

Yes

Are the conclusions supported by the results?

Yes

3. Point-by-point response to Comments and Suggestions for Authors

Comment 1: The authors need to add the details of statistical tests in Figures 1–4 to their legends.

Response 1: We thank Reviewer for this comment. According to Reviewer’s request, we have added the following text in the legends of Figures 1-4: “The statistical analysis of the results was performed through two-way ANOVA with repeated measures”.

Comment 2: In addition, they must describe the procedure of their statistical analysis of this study to the experimental section of the revised manuscript.

Response 2: We would like to inform Reviewer that we have clarified the statistical analysis applied in the study. In particular, we have added the following text in the Section 2.7 “Statistical analysis” of the revised manuscript: “The alterations of the levels of craving as well as the concentrations of the biomarkers measured in blood were assessed through two-way, 2 x 2 (i.e., MAT dose x time) analysis of variance (ANOVA) with repeated measures”.
